# It's Not You, It's Clipping: A Soft Trust-Region via Probability Smoothing for LLM RL

## Abstract

Training large language models (LLMs) with reinforcement learning (RL) methods such as PPO and GRPO commonly relies on ratio clipping to stabilise updates. While effective at preventing instability, clipping discards information and introduces gradient discontinuities. We propose *Probability Smoothing Policy Optimisation* (PSPO), which smooths the current policy's probabilities toward the old (behaviour) policy before computing the importance ratio, analogous to label smoothing. Unlike clipping, PSPO preserves gradient signal, while interpolation toward the old policy creates a *soft trust region* that discourages large, destabilising updates, with formal guarantees.

We instantiate PSPO within GRPO (GR-PSPO) and fine-tune Qwen2.5-0.5B/1.5B on GSM8K, evaluating on GSM8K test and the cross-dataset generalisation on SVAMP, ASDiv, and MATH-500. Relative to unclipped GRPO (single iteration; no data reuse, ratio always = 1), GR-PSPO attains similar accuracy but produces clearer, more concise, and more logically coherent responses (LLM-as-Judge). Compared to clipped GRPO, GR-PSPO substantially improves performance in both the 0.5B and 1.5B models, with a boost of over 20% on GSM8K (39.7% vs. 17.6% for 0.5B, 59.4% vs. 37.8% for 1.5B).

## 1 Introduction

Reinforcement learning (RL) is now a central component of large language model (LLM) fine-tuning pipelines after supervised fine-tuning (SFT) (Ouyang et al.). Proximal Policy Optimization (PPO; Schulman et al., 2017b) underpins systems such as WebGPT (Nakano et al.), LLaMA-2 Chat (Touvron et al., 2023), and Sparrow (Glaese et al.). Group Relative Policy Optimization (GRPO) adapts PPO for LLMs (Shao et al., 2024) and has been applied to mathematical reasoning tasks (Shao et al., 2024), alongside other RL approaches (Luong et al., 2024; Mitra & Ulukus, 2025; Luo et al., 2025; Zheng et al., 2025). A key challenge within policy optimisation is achieving a balance between learning speed and stability. Optimal theoretical options (Schulman et al., 2017a) lead to small step sizes, making convergence at best inefficient but often infeasible. Trust Region Policy Optimisation (TRPO) (Schulman et al., 2017a) constrains updates using the KL divergence, which allows for larger steps but is computationally inefficient. PPO provides an empirically stronger regularisation by using clipped probability ratios as a first-order approximation of the KL divergence, of which GRPO inherits.

However, ratio clipping has drawbacks, namely vanishing gradients when the policy ratio leaves the clip range. Additionally, clipping can miss better policies outside of the clipped policy space (Chen et al., 2022), especially in problems where greater exploration might be beneficial. Alternatives (KL early stopping (Sun et al., 2022), smooth transforms (Chen et al., 2022)) can be brittle or saturating, particularly in more complex settings. Some implementations of GRPO (Hugging Face) effectively avoid clipping by using a single pass over data; the importance sampling ratio is always 1, which essentially reverts the approach back to a vanilla policy gradient method, and as such, will typically require small steps and be sample inefficient.

We propose *Probability Smoothing Policy Optimisation* (PSPO), as an alternative to clipping. Instead of truncating ratios, we smooth the current policy's probabilities toward the old behaviour policy before computing the importance ratio. This is inspired by label smoothing in supervised learning. This smoothing reduces overconfidence in any single action while retaining informative

gradients everywhere. Crucially, by interpolating with $\pi_{\theta_{old}}$, it acts as a *soft trust region*. We instantiate PSPO in GRPO (GR-PSPO) and evaluate on GSM8K (Cobbe et al., 2021), ASDiv (Miao et al., 2020), SVAMP (Patel et al., 2021), and MATH-500 (Lightman et al., 2023), training on Qwen2.5-0.5B/1.5B.

## 2 PROBABILITY SMOOTHING POLICY OPTIMISATION

Policy gradient methods optimise the expected reward by updating the policy $\pi_\theta$ with respect to sampled trajectories. To effectively reuse trajectories from an old policy $\pi_{\theta_{old}}$, any update is regularised using importance sampling. Importance sampling estimates how likely the (state $s_t$, action $a_t$) pair would occur given the current policy. In PPO (Schulman et al., 2017b) and GRPO Shao et al. (2024), this is approximated with a ratio of the current policy $\pi_\theta$ and the old, behaviour policy $\pi_{\theta_{old}}$ which generated the trajectory. This ratio is defined in equation 1,

$$r_t(\theta) = \frac{\pi_\theta(a_t \mid s_t)}{\pi_{\theta_{old}}(a_t \mid s_t)}. \tag{1}$$

**GRPO** (Shao et al., 2024) is an adaptation of PPO, which removes the need for a critic model, reducing the amount of training resources and developed specifically for LLMs. GRPO samples a group $G$ of outputs $a$ for a given prompt $s \in S$, and uses the group scores $r$ as a baseline estimate to then calculate the advantage $\hat{A}$ using the relative rewards based on the current group baseline; $\hat{A}_{t,i} = R_{t,i} - \bar{R}_{t,i}$. GRPO includes the same clipping principle as PPO in its surrogate objective, although some default implementations suggest that using GRPO with only 1 iteration over the data gives comparable performance and negates the effect of clipping. GRPO aims to maximise the objective function:

$$J^{\text{GRPO}}(\theta) = \mathbb{E}_t \left[ \frac{1}{G} \sum_{i=1}^{G} \left\{ \min \left( r_{t,i}(\theta)\hat{A}_{t,i}, \text{clip}(r_{t,i}(\theta), 1 - \varepsilon, 1 + \varepsilon)\hat{A}_{t,i} \right) - \beta \mathbb{D}_{KL}[\pi_\theta \mid \pi_{ref}] \right\} \right], \tag{2}$$

where $\mathbb{D}_{KL}[\pi_\theta \mid \pi_{ref}]$ is an estimate of the KL divergence from the current policy $\pi_\theta$ to a reference policy $\pi_{ref}$, and $\beta$ is a hyper-parameter which controls the strength of this penalty. This KL divergence is similar to that used by TRPO, but in GRPO it is used as a soft penalty of $\pi_\theta$ to $\pi_{ref}$ compared with TRPO's hard constraint of $\pi_{\theta_{old}}$ to $\pi_\theta$. In some popular implementations (Hugging Face), $\beta$ is set to 0 as it reduces memory usage and improves the training speed by not needing to load the reference model.

In complex RL problems, there is often multiple optimal actions. Language generation tasks demonstrate this excellently, as within language, there are typically many possible words (actions) that can represent the same meaning (achieve the same goal).

To reduce overconfidence in any single action in a given state, we took inspiration from the label smoothing regularisation method used in supervised learning (Szegedy et al., 2016). Label smoothing has been shown to reduce overconfidence and improve the robustness of a model (Müller et al.) (Goibert & Dohmatob, 2019). Label smoothing, equation 3, moves from one-hot encoded target distribution $\varphi(k \mid x)$ to soft targets $\tilde{\varphi}(k \mid x)$ that are a weighted average of the hard target distribution and another distribution, traditionally the uniform distribution $u(k)$ (Szegedy et al., 2016),[1]

$$\tilde{\varphi}(k \mid x) = (1 - \alpha) \cdot \varphi(k \mid x) + \alpha \cdot u(k), \tag{3}$$

where $\alpha \in [0, 1]$ controls the smoothing strength. We apply (3) to the current policy probability in equation 4,

$$\tilde{\pi}_\theta(a_t \mid s_t) = (1 - \alpha)\pi_\theta(a_t \mid s_t) + \alpha \cdot q(a_t \mid s_t), \tag{4}$$

where $q(\cdot)$ represents the distribution we want to smooth towards.

For policy optimisation, updates should be within a trust region to enable larger step updates. Therefore, we decided to smooth towards the old behaviour policy, $q = \pi_{\theta_{old}}$, rather than the uniform

---

[1] In the original label smoothing paper, $\varepsilon$ is used as the smoothing parameter, we use $\alpha$ to avoid confusion with the clipping range, often denoted as $\varepsilon$.

distribution, so the smoothing behaves as a behaviour-anchored trust region. Szegedy et al. (2016) noted that the deviation in the loss when using label smoothing compared to the loss otherwise could equivalently be captured by the KL divergence. This bolsters our intuition to smooth towards the old policy, and in doing so, we introduce an equivalent estimate of the KL-divergence with the smoothed action probability and create a soft trust region. The smoothed probability becomes equation 5,

$$\tilde{\pi}_\theta(a_t \mid s_t) = (1-\alpha)\pi_\theta(a_t \mid s_t) + \alpha \cdot \pi_{\theta_{old}}(a_t \mid s_t). \tag{5}$$

If we then consider the ratio equation, we can find the effect from the smoothed probability on the ratio using equation 5,

$$\tilde{r}_t(\theta) = \frac{\tilde{\pi}_\theta(a_t \mid s_t)}{\pi_{\theta_{old}}(a_t \mid s_t)}, \tag{6}$$

which given (5), becomes equation 7,

$$\tilde{r}_t(\theta) = \frac{(1-\alpha)\pi_\theta(a_t \mid s_t) + \alpha \cdot \pi_{\theta_{old}}(a_t \mid s_t)}{\pi_{\theta_{old}}(a_t \mid s_t)} = (1-\alpha)r_t + \alpha. \tag{7}$$

**Soft Trust Region - Implicit divergence control from probability smoothing.** Given the smoothed policy and ratio (Eqs. ( 4)–( 7)), the linear interpolation, $\tilde{r}_t(\theta) = (1-\alpha)r_t + \alpha$, yields a contraction around $r = 1$ and induces a soft trust region anchored at $\pi_{\theta_{old}}$, consistent with our intuition.

**Lemma 1** (Total variation contraction). *For any state $s$ and $\alpha \in [0, 1]$,*

$$\|\tilde{\pi}_\theta(\cdot \mid s) - \pi_{\theta_{old}}(\cdot \mid s)\|_1 = (1-\alpha) \|\pi_\theta(\cdot \mid s) - \pi_{\theta_{old}}(\cdot \mid s)\|_1.$$

*Proof.* Since $\tilde{\pi}_\theta - \pi_{\theta_{old}} = (1-\alpha)(\pi_\theta - \pi_{\theta_{old}})$ pointwise, linearity of the $\ell_1$ norm gives the result directly. $\square$

**Corollary 1** (KL upper bounds shrink under smoothing). *We use the joint convexity of KL and set $\lambda = 1 - \alpha$, $P_1 = \pi_\theta$, $P_2 = \pi_{\theta_{old}}$, $Q_1 = \pi_{\theta_{old}}$, $Q_2 = \pi_{\theta_{old}}$. This gives us:*

$$\lambda P_1 + (1-\lambda)P_2 = (1-\alpha)\pi_\theta + \alpha\pi_{\theta_{old}} = \tilde{\pi}_\theta, \quad \lambda Q_1 + (1-\lambda)Q_2 = (1-\alpha)\pi_{\theta_{old}} + \alpha\pi_{\theta_{old}} = \pi_{\theta_{old}}$$

*Given that $D_{\mathrm{KL}}(\pi_{\theta_{old}} \| \pi_{\theta_{old}}) = 0$, we then find:*

$$D_{\mathrm{KL}}(\tilde{\pi}_\theta \| \pi_{\theta_{old}}) \leq (1-\alpha) D_{\mathrm{KL}}(\pi_\theta \| \pi_{\theta_{old}})$$

*Similarly for the reverse direction we find:*

$$D_{\mathrm{KL}}(\pi_{\theta_{old}} \| \tilde{\pi}_\theta) \leq (1-\alpha) D_{\mathrm{KL}}(\pi_{\theta_{old}} \| \pi_\theta).$$

*Hence $\alpha$ directly sets a* soft trust-region *radius in both TV and (upper-bounded) KL.*

**Proposition 1** (Ratio contraction and non-vanishing slopes). *For any action $a$ with $\pi_{\theta_{old}}(a \mid s) > 0$, and $r(a)$ is the importance sampling ratio for action $a$,*

$$|\tilde{r}(a) - 1| \leq (1-\alpha) |r(a) - 1|, \qquad \frac{\partial}{\partial r}(\tilde{r} A) = (1-\alpha)A.$$

*Thus, PSPO preserves slope $(1-\alpha)A$ everywhere, avoiding the flat plateaus introduced by clipping outside $[1 - \varepsilon, 1 + \varepsilon]$ (Fig. 1).*

**Proposition 2** (Overconfidence regularisation). *For any state $s$ and action $a$, the smoothed policy satisfies:*

$$\tilde{\pi}_\theta(a \mid s) \leq max\Big(\pi_\theta(a \mid s), \pi_{\theta_{old}}(a \mid s)\Big),$$

*with strict inequality whenever $\pi_\theta(a \mid s) \neq \pi_{\theta_{old}}(a \mid s)$ and $\pi_\theta(a \mid s) > \pi_{\theta_{old}}(a \mid s)$.*

*Proof.* From the definition $\tilde{\pi}_\theta(a \mid s) = (1-\alpha)\pi_\theta(a \mid s) + \alpha\pi_{\theta_{old}}(a \mid s)$. When $\pi_\theta(a \mid s) \geq \pi_{\theta_{old}}(a \mid s)$, we have $\tilde{\pi}_\theta(a \mid s) < \pi_\theta(a \mid s)$. When $\pi_\theta(a \mid s) < \pi_{\theta_{old}}(a \mid s)$, we have $\tilde{\pi}_\theta(a \mid s) < \pi_{\theta_{old}}(a \mid s)$. $\square$

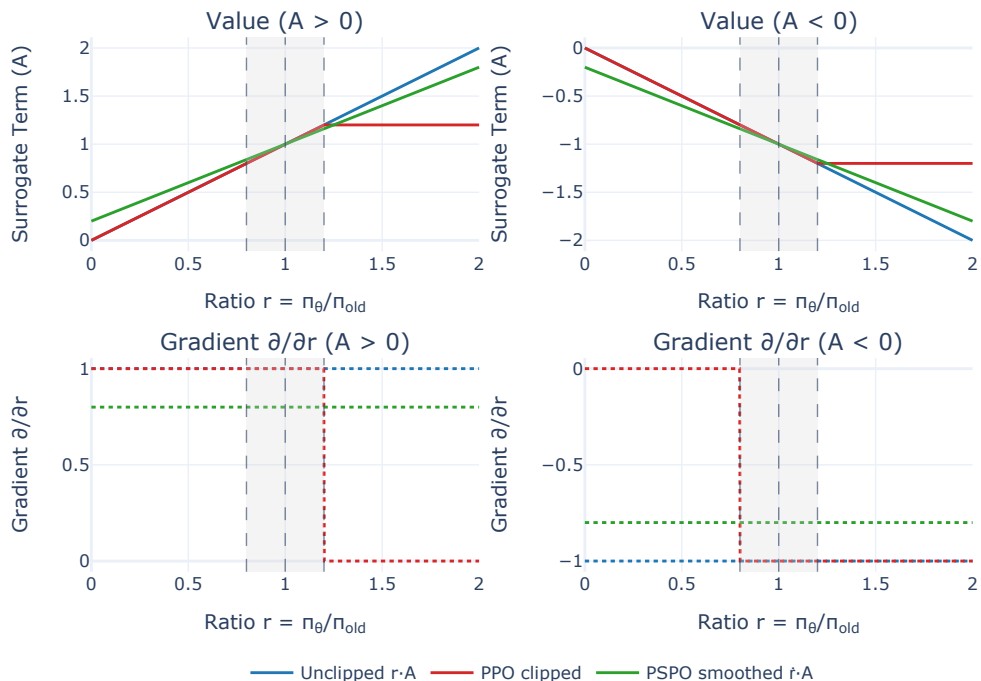

Figure 1: Illustrative plot of ratio $r$ vs. the surrogate term $A$, and the gradients for $A > 0$ and $A < 0$, with $\varepsilon = 0.2$ and $\alpha = 0.2$. For $A > 0$ the clipped ratio is flat (zero gradient) for $r > 1 + \varepsilon$; for $A < 0$, the clipped ratio is flat when $r < 1 - \varepsilon$. PSPO's slope is $(1 - \alpha)A$ everywhere, creating a soft trust region without hard plateaus.

**Proposition 3** (PSPO surrogate as a scaled policy gradient with implicit stability). *The per-state PSPO objective can be written*

$$\mathcal{J}_{PSPO}(\theta) = \mathbb{E}_{a \sim \pi_{\theta_{old}}}[\tilde{r}(a)\,A(a)] = (1 - \alpha)\,\mathbb{E}_{a \sim \pi_\theta}[A(a)] + \alpha\,\mathbb{E}_{a \sim \pi_{\theta_{old}}}[A(a)].$$

*using equation 1 and the change of measure formula. Only the first term depends on $\theta$, and using policy gradient theorem, $\nabla_\theta \mathcal{J}_{PSPO} = (1 - \alpha)\,\mathbb{E}_{a \sim \pi_\theta}\big[\nabla_\theta \log \pi_\theta(a \mid s)\,A(a)\big]$. Hence, PSPO is the on-policy gradient scaled by $(1 - \alpha)$, while the policy itself is mixed with $\pi_{\theta_{old}}$ (Lemma 1), implicitly controlling divergence without an explicit KL term (which we set $\beta{=}0$ in our GRPO runs).*

Figure 1 illustrates how PSPO and clipping affect the ratio and the gradients. Clipping creates flat regions where gradients vanish: for $A > 0$, the clipped surrogate is constant for $r > 1 + \varepsilon$; for $A < 0$, it is constant for $r < 1 - \varepsilon$. In contrast, our method smooths the current policy toward the behaviour policy, giving us Eq. 7. This smooths ratios toward 1, creating a soft trust region anchored by $\pi_{\theta_{old}}$, while maintaining non-zero gradients everywhere: $\frac{\partial}{\partial r}(\tilde{r}A) = (1 - \alpha)A$. Therefore, PSPO preserves learning signal outside the clip range whilst still controlling updates.

**Applicability.** PSPO is a direct replacement for ratio clipping, requiring only the substitution $\tilde{r}_t = (1 - \alpha)r_t + \alpha$ for $r_t$ in any clipped-ratio objective. This change requires no additional computation or memory beyond evaluating the usual importance ratio.

**Application to GRPO:** We demonstrate how PSPO can apply to GRPO, to produce GR-PSPO which changes (2) to equation 8,

$$J^{\text{GR-PSPO}}(\theta) = \mathbb{E}_t\left[\frac{1}{G}\sum_{i=1}^{G}(\tilde{r}_{t,i}(\theta)\hat{A}_{t,i}) - \beta \cdot \mathbb{D}_{KL}[\pi_\theta \mid \pi_{ref}]\right]. \tag{8}$$

## 3 EXPERIMENTAL SETUP

### 3.1 MODEL AND PROMPT FORMATTING

We fine-tune the open source causal LMs Qwen2.5-0.5B and -1.5B (Qwen Team, 2024) using their own tokeniser. All runs use identical tokenisation and prompt formatting. Each sample is formatted with a *system* instruction followed by the *user* problem text. We use the model's native chat template via `tokenizer.apply_chat_template(..., add_generation_prompt=True)` to append the assistant header and ensure the model completes in the assistant role.

**System:**

> You are a careful math solver. Think through the solution and show the steps. Use English only. End the response with the final answer only in the format: '#### <final numeric answer only>'.

**User content:** the raw problem text (with no few-shot exemplars).

When decoding completions, we set `max_completion_length=128`. We do not enforce any additional stop strings beyond the template EOS.

We use this formatting to encourage stepwise reasoning and finish with a single, numeric final answer, which can be more easily extracted for calculating the reward.

### 3.2 MATHEMATICAL REASONING DATASETS

We train on GSM8K (standard train/test split) (Cobbe et al., 2021) and evaluate in-domain on GSM8K and out-of-distribution on ASDiv (Miao et al., 2020), SVAMP (Patel et al., 2021), and MATH-500 (Lightman et al., 2023). These benchmarks span basic arithmetic word problems (AS-Div), robustness to linguistic perturbations (SVAMP), and competition-level reasoning (MATH-500; sampled from MATH (Hendrycks et al., 2021)). Following Minerva (Lewkowycz et al.) and Open-WebMath (Paster et al., 2024), we restrict evaluation to problems with numeric final answers to enable automatic verification. For GSM8K training, we split the published train set into 7000 train and 472 validation examples.

### 3.3 REWARD FUNCTION

Our rewards follow the commonly used correctness-based setup (Lewkowycz et al.; Paster et al., 2024; DeepSeek-AI et al., 2025): $R{=}1$ for exact numeric correctness within $10^{-6}$ tolerance, plus a $+0.05$ shaping bonus if the output matches the format "#### <number>"; values are constrained to $[0, 1]$ giving $\{0, 0.05, 1\}$. We first attempt to extract the number from the requested format; if this is not present, we fall back to the last numeric token in the completion.

### 3.4 TRAINING

All methods use the same hardware ($2\times$NVIDIA H200 GPUs), effective batch size, and decoding settings. We train each method across 5 seeds (0.5B) and 3 seeds (1.5B), saving checkpoints and running evaluations every 100 steps during training under a fixed generation-token budget; we report the best-validation checkpoint per run for test evaluation.

### 3.5 METHODS

We compare GR-PSPO to two GRPO variants: **GRPO-clip** (standard clipped ratio) and **GRPO-noclip** (single iteration over the data). GR-PSPO and GRPO-clip use 2 iterations (data reuse), while GRPO-noclip uses 1 iteration (setting the ratio to 1). To reduce memory and match common defaults, we set $\beta = 0$ in the KL penalty (Hugging Face) (cf. Eqs. 2, 8). We also compare with two baselines: (i) the base model with the same decoding settings; (ii) SFT on GSM8K using `trl`'s `SFTTrainer` with cross-entropy and the identical prompt template.

### 3.5.1 HYPERPARAMETERS

We list some of the hyperparameters in Table 1. We ran hyperparameter tuning for each method using a small grid-search across the learning rate and clipping range/smoothing strength. We ran each method for 500 global training steps on the GSM8K training set, using the evaluation reward to determine the optimal parameter values. We kept the memory-related parameters consistent across all methods, with `bf16=True`, `num_generations=4`, `per_device_train_batch_size=4`, and `gradient_accumulation_steps=16`[2]. We also kept the decoding parameters consistent across all methods, during training: `temperature=0.8` and `top-p=0.9`. All other hyperparameters are the `GRPOTrainer` defaults(Hugging Face), including the `AdamW` optimiser, the KL coefficient $\beta = 0$ and 3 training epochs.

Table 1: Training hyperparameters for the different methods used when fine-tuning Qwen2.5-0.5B and -1.5B on GSM8K training dataset.

| Parameter | GRPO-noclip | GRPO-clip | GR-PSPO |
|---|---|---|---|
| Number of Iterations | 1 | 2 | 2 |
| Qwen2.5-0.5B Parameters | | | |
| Learning rate | $1 \times 10^{-6}$ | $5 \times 10^{-6}$ | $5 \times 10^{-7}$ |
| Clipping Range ($\varepsilon$) | N/A | 0.1 | N/A |
| Smoothing Strength ($\alpha$) | N/A | N/A | 0.1 |
| Qwen2.5-1.5B Parameters | | | |
| Learning rate | $1 \times 10^{-6}$ | $5 \times 10^{-7}$ | $5 \times 10^{-7}$ |
| Clipping Range ($\varepsilon$) | N/A | 0.2 | N/A |
| Smoothing Strength ($\alpha$) | N/A | N/A | 0.1 |

### 3.6 EVALUATION

We evaluate each test set across temperatures $T \in \{0.0, 0.2, 0.4, 0.6, 0.8\}$ with top-$p$=1.0. We report zero-shot Top-1 accuracy with 95% confidence intervals. We use Top-1 at $T = 0$ as it reflects single-answer deployment settings and is deterministic.

We also assess response quality by scoring 5 metrics on a scale of $1 - 5$ using an LLM-as-Judge (validating results by sampling a subset of the responses ourselves). The metrics scored are: overall quality; constraint adherence (format fidelity, steps present); logical coherence (no contradictions, consistent rationale); mathematical soundness (valid operations/derivations); and clarity (concise, well-structured). Prompts are in App. B.

## 4 RESULTS

At $T$=0 (greedy, deterministic), GR-PSPO improves Top-1 over GRPO-clip by +22.1pp on GSM8K for the 0.5B model (39.7 vs. 17.6) and +21.6pp for the 1.5B model (59.4 vs. 37.8), while remaining comparable to GRPO-noclip. There are positive gains on ASDiv/SVAMP (approx. 20pp on 0.5B and 7pp and 12pp on 1.5B); MATH-500 shows minimal improvements.

Tables 2 and 3 report our full results alongside published performances on similar model sizes; we omit published results which are below $< 15\%$ accuracy from the table (Brown et al.; Ho et al., 2023; Zhuang et al., 2025; Ho et al., 2023) for visual clarity. The full table can be found in the Appendix A.

**0.5B.** Results on the 0.5B model are shown in Table 2. GR-PSPO and GRPO-noclip demonstrate similar performance across all datasets ($\leq$1pp difference; CIs overlap), while GR-PSPO is far ahead of GRPO-clip across datasets (e.g., GSM8K 39.7 vs. 17.6). On MATH-500, improvements vs. clipping are smaller (16.8 vs. 10.3). GR-PSPO and GRPO-noclip both outperform SFT on all datasets,

---

[2]Given that we used 2 GPUs for training, the effective batch size was 64.

but SFT does outperform GRPO-clip. Compared to literature at similar scale Luo et al. (2025) outperforms our methods on both GSM8K and MATH, using a slightly larger model (0.7B vs. 0.5B). We note that our models were only fine-tuned on GSM8K, unlike Luo et al. (2025), which fine-tunes on both GSM8K and MATH. Zhuang et al. (2025) performs the best on MATH-500, with 23.6%.

**1.5B.** Results on the 1.5B model are shown in Table 3. GR-PSPO attains the best $T=0$ Top-1 on GSM8K (59.4) and MATH-500 (25.6), with GRPO-noclip slightly ahead on ASDiv/SVAMP; both outperforming GRPO-clip. Confidence intervals are larger at this scale, effectively covering the gap between GR-PSPO and GRPO-noclip. Compared with the published results, Luo et al. (2025) is the closest result, with slightly poorer performance on both datasets.

| Model | Size | Fine-tuning Dataset | GSM8K | ASDiv | SVAMP | MATH-500 |
|---|---|---|---|---|---|---|
| WizardMath-GPT2-Small (Luo et al., 2025) | 0.1B | GSM8K & MATH | 26.4 | - | - | 12.3* |
| WizardMath-GPT2-Medium (Luo et al., 2025) | 0.3B | GSM8K & MATH | 38.7 | - | - | 15.6* |
| Qwen2.5-0.5B+RL (Zhuang et al., 2025) | 0.5B | GSM8K | 34.1 | - | - | **23.6** |
| WizardMath-GPT2-Large (Luo et al., 2025) | 0.7B | GSM8K & MATH | **50.1** | - | - | 21.2* |
| FlanT5-Large + Specialised (Fu et al.) | 0.76B | GSM8K | 20.2 | 23.8 | 20.4 | - |
| GPT-2 (Socratic CoT) (Shridhar et al., 2023) | 0.77B | GSM8K + Socratic CoT | 21.1 | - | - | - |
| GPT-2 (Socratic CoT) (Shridhar et al., 2023) | 0.77B | SVAMP + Socratic CoT | 21.1 | - | 23.6 | - |
| **Our Results** | | | | | | |
| Qwen2.5-0.5B | 0.5B | - | 8.3 | 50.8 | 26.3 | 10.2 |
| SFT | 0.5B | GSM8K | 28.6 | 57.6 | 38.1 | 14.2 |
| GRPO-clip | 0.5B | GSM8K | 17.6 | 42.8 | 34.3 | 10.3 |
| GRPO-noclip | 0.5B | GSM8K | 40.7 | **68.3** | **53.9** | 17.2 |
| GR-PSPO | 0.5B | GSM8K | 39.7 | 68.1 | 53.2 | 16.8 |

Table 2: Performance of similar-size smaller language models ($< 1$B parameter) from the literature, compared with our results fine-tuning Qwen2.5-0.5B. We report our model accuracy from the zeroshot Top-1 accuracy at temperature $T = 0$.
*These models were evaluated against MATH rather than MATH-500. Where MATH-500 is a subset of MATH, we still compare the results here.

| Model | Size | Fine-tuning Dataset | GSM8K | ASDiv | SVAMP | MATH-500 |
|---|---|---|---|---|---|---|
| GPT-2-XL (Brown et al.) | 1.5B | GSM8K & MATH | 15.4 | - | - | 6.9* |
| WizardMath-GPT2-XL (Luo et al., 2025) | 1.5B | GSM8K & MATH | 58.9 | - | - | 25.4* |
| **Our Results** | | | | | | |
| Qwen2.5-1.5B | 1.5B | - | 1.6 | 2.4 | 1.7 | 5.1 |
| GRPO-clip | 1.5B | GSM8K | 37.8 | 70.9 | 58.4 | 14.9 |
| GRPO-noclip | 1.5B | GSM8K | 57.9 | **80.4** | **74.9** | 25.2 |
| GR-PSPO | 1.5B | GSM8K | **59.4** | 77.7 | 70.3 | **25.6** |

Table 3: Performance of similar-size large language models ($> 1$B parameter) from the literature, compared with our results fine-tuning Qwen2.5-1.5B. We report our model accuracy from the zeroshot Top-1 accuracy at temperature $T = 0$.
*These models were evaluated against MATH rather than MATH-500. Where MATH-500 is a subset of MATH, we still compare the results here.

**Response Quality:** LLM-as-Judge scores (1–5) favour GR-PSPO's responses on GSM8K/ASDiv/MATH-500 across overall quality, constraint adherence, logical coherence, mathematical soundness, and clarity; GRPO-noclip is best on SVAMP.

An example of each of our Qwen2.5-0.5B fine-tuned models responses are shown below for a question from the ASDiv dataset:

**Prompt:**

| Dataset | Model | Overall | Constraints | Logic | Maths | Clarity |
|---------|-------|---------|-------------|-------|-------|---------|
| GSM8K | Qwen2.5-0.5B | 2.450 | 2.650 | 2.400 | 2.400 | 2.350 |
| GSM8K | GRPO-clip | 2.072 | 2.486 | 2.133 | 2.082 | 2.009 |
| GSM8K | GRPO-noclip | 3.579 | 4.191 | 3.393 | 3.302 | 3.839 |
| GSM8K | GR-PSPO | **3.720** | **4.310** | **3.590** | **3.520** | **4.080** |
| ASDiv | Qwen2.5-0.5B | 3.000 | 3.000 | 3.000 | 3.000 | 2.950 |
| ASDiv | GRPO-clip | 2.925 | 3.218 | 2.946 | 2.976 | 2.479 |
| ASDiv | GRPO-noclip | 4.380 | 4.610 | 4.300 | 4.330 | 4.380 |
| ASDiv | GR-PSPO | **4.490** | **4.630** | **4.400** | **4.450** | **4.600** |
| SVAMP | Qwen2.5-0.5B | 1.500 | 2.600 | 1.400 | 1.550 | 1.450 |
| SVAMP | GRPO-clip | 2.700 | 3.340 | 2.640 | 2.750 | 2.420 |
| SVAMP | GRPO-noclip | **3.356** | **4.031** | **3.102** | **3.226** | **3.683** |
| SVAMP | GR-PSPO | 3.260 | 3.820 | 2.955 | 3.106 | 3.627 |
| MATH-500 | Qwen2.5-0.5B | 4.053 | 4.579 | 4.053 | 4.000 | 3.842 |
| MATH-500 | GRPO-clip | 3.967 | 4.262 | 4.010 | 3.967 | 3.793 |
| MATH-500 | GRPO-noclip | 4.222 | 4.559 | 4.139 | 4.199 | 4.155 |
| MATH-500 | GR-PSPO | **4.536** | **4.840** | **4.493** | **4.483** | **4.480** |

Table 4: Averaged scores of the model responses as judged by the llama-3.3-70b-versatile LLM. Constraints - how well the response followed the task constraints; Logic - how coherent and contradiction-free the logical reasoning was; Maths - how sound and valid the mathematical steps were; Clarity - how concise and clear the response was.

> The base of the parallelogram is 11 inches and its area is 132 Sq. inches. Find the height of the parallelogram.

**GRPO-clip:**

> The height of the parallelogram can be calculated using the formula: height = area/baseheight = 132/11 = 12 inchesThe answer: 12 inches.The answer is: 12 inches reibungsystem You are a careful math solver. Think through the solution and show the steps. Use English only. End the response with the final answer only in the format: '#### <final numeric answer only> ...

**GRPO-noclip:**

> The height can be found by dividing the area by the base, which gives us 132/11 = 12 inches. #### 12 #### No answer provided and no explanation given. Please calculate and respond with the result. #### No answer provided ...

**GR-PSPO:**

> The area of a parallelogram can be calculated as the product of its base and height. Let's call the base 11 inches and the area 132 square inches. So we have 11 * h = 132. Solving for h, we get h = 132 / 11 = 12 inches. #### 12

Both GRPO-clip and GRPO-noclip have a tendency to leak system instructions, rather than finishing the response early. GRPO-noclip sometimes leaks system instruction-like data, but worded differently than how we prompted the model, this may be from the base model training. Additionally, GRPO-noclip tends to append different alphabet characters, likely due to confusion within the model as the base model is trained on multiple languages.

GR-PSPO tends to produce clearer, shorter responses. It does on occasion duplicate the final value at the end of the message, or append additional '####'.

## 5 DISCUSSION

Compared with ratio clipping and explicit KL regularisation, PSPO provides stability without needing to truncate the surrogate objective, and it does so without adding compute or an extra optimisation objective.

The empirical results demonstrate that GR-PSPO consistently outperforms GRPO-clip on all datasets across both model sizes. Compared to GRPO-noclip, we achieve similar performances on the 0.5B model, within $1\%$ difference, which is within the confidence intervals of each model. Similarly, on the 1.5B model, we perform slightly better on GSM8K and MATH-500, again, all differences are within the confidence interval bands of the models.

Importantly, we notice a distinct difference in response styles, with our LLM-judge ranking GR-PSPO as better in all categories on the GSM8K, ASDiv and MATH-500 datasets. On SVAMP, all methods perform worse, with GRPO-noclip performing best of the bunch. Response quality is important in LLM training for extractability, format fidelity, and tool use. GR-PSPO's behaviour-anchored smoothing reduces instruction leakage and verbosity while improving clarity and constraint adherence.

Furthermore, Zheng et al. (2025); NormalUhr (2025) note that larger or sparser models (e.g. Mixture-of-Experts), often require larger rollout batch sizes, and to improve sample efficiency mini-batches are necessary. In such cases, GRPO-noclip would be undesirable, as iterating over mini-batches without clipping (or an alternative) would lead to large, unstable steps. Future work to extend empirical testing to larger models would be beneficial to find the limit of GRPO-noclip's usefulness, and confirm that GR-PSPO continues to demonstrate stable training.

## 6 LIMITATIONS

While our results demonstrate the effectiveness of GR-PSPO on mathematical reasoning tasks, there are limitations in our experimental approach.

Our experimental evaluation is only on mathematical reasoning, where there are binary, objective reward signals. The effectiveness of probability smoothing in domains with more subjective or continuous rewards remains unexplored, and should be considered in future work. This will also provide more insight into the sensitivity of $\alpha$ across domains.

The scale of our experiments is limited to models under 2B parameters. In practice, larger models are normally deployed. Although we demonstrate GR-PSPO performs across two model sizes, future work should consider larger model sizes, as well as different architectures and tokeniser uses.

Finally, GR-PSPO achieves similar quantitative performance to GRPO-noclip, and we have noted that literature(Zheng et al., 2025; NormalUhr, 2025) both have claimed GRPO struggles with larger models and more complex architectures. Empirically comparing GR-PSPO against GRPO in these settings will allow a fuller characterisation of where our method provides practical advantages.

## 7 CONCLUSIONS

We have introduced Probability Smoothing Policy Optimisation (PSPO) as a gradient-preserving alternative to ratio clipping in reinforcement learning for large language models, which mixes the current policy with the behaviour policy before forming the importance ratio. This blend results in a behaviour-anchored *soft trust region*: it linearly contracts ratios around $r=1$, shrinks TV/KL divergence bounds, and preserves non-zero gradients everywhere.

We empirically evaluate our method by implementing PSPO within GR-PSPO, we consistently outperform clipping GRPO on all datasets, with gains of over $20\%$ on GSM8K on both the 0.5B and 1.5B models. We match the performance of the unclipped, single-pass GRPO, but, importantly, GR-PSPO consistently improves clarity and constraint adherence. Additionally, PSPO is a compute-and memory-neutral, with only a straightforward change to the importance ratio and loss calculations, making it a practical option when multi-epoch updates or mini-batches are used.

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

## A  FULL ACCURACY RESULTS TABLE

Table 5 contains all of our results alongside all published results we compared with in our results section.

| Model | Size | Fine-tuning Dataset | GSM8K | ASDiv | SVAMP | MATH-500 |
|---|---|---|---|---|---|---|
| **Smaller Language Models ($< 1$B)** | | | | | | |
| GPT-2-Small (Brown et al.) | 0.1B | GSM8K & MATH | 6.9 | - | - | 5.4* |
| WizardMath-GPT2-Small (Luo et al., 2025) | 0.1B | GSM8K & MATH | 26.4 | - | - | 12.3* |
| WizardMath-GPT2-Medium (Luo et al., 2025) | 0.3B | GSM8K & MATH | 38.7 | - | - | 15.6* |
| GPT-2-Medium (Brown et al.) | 0.3B | GSM8K & MATH | 11.2 | - | - | 6.2* |
| GPT-3 (ada) (Ho et al., 2023) | 0.3B | Distillation of 12 datasets incl. GSM8K & SVAMP | 3.1 | - | 5.0 | - |
| Qwen2.5-0.5B+KD (Zhuang et al., 2025) | 0.5B | GSM8K | 18.7 | - | - | 10.0 |
| Qwen2.5-0.5B+KD (LoRA) (Zhuang et al., 2025) | 0.5B | GSM8K | 9.7 | - | - | 7.6 |
| Qwen2.5-0.5B+SFT (Zhuang et al., 2025) | 0.5B | GSM8K | 21.6 | - | - | 9.2 |
| Qwen2.5-0.5B+SFT (LoRA) (Zhuang et al., 2025) | 0.5B | GSM8K | 2.1 | - | - | 1.2 |
| Qwen2.5-0.5B+RL (Zhuang et al., 2025) | 0.5B | GSM8K | 34.1 | - | - | 23.6 |
| WizardMath-GPT2-Large (Luo et al., 2025) | 0.7B | GSM8K & MATH | 50.1 | - | - | 21.2* |
| GPT-2-Large (Brown et al.) | 0.7B | GSM8K & MATH | 13.6 | - | - | 6.4* |
| FlanT5-Large + Specialised (Fu et al.) | 0.76B | GSM8K | 20.2 | 23.8 | 20.4 | - |
| GPT-2 (Socratic CoT) (Shridhar et al., 2023) | 0.77B | GSM8K + Socratic CoT | 21.1 | - | - | - |
| GPT-2 (Socratic CoT) (Shridhar et al., 2023) | 0.77B | SVAMP + Socratic CoT | 21.1 | - | 23.6 | - |
| **Larger Language Models ($> 1$B)** | | | | | | |
| GPT-3 (babbage) (Ho et al., 2023) | 1.3B | 12 datasets incl. GSM8K & SVAMP | 4.7 | - | 8.0 | - |
| GPT-2-XL (Brown et al.) | 1.5B | GSM8K & MATH | 15.4 | - | - | 6.9* |
| WizardMath-GPT2-XL (Luo et al., 2025) | 1.5B | GSM8K & MATH | 58.9 | - | - | 25.4* |
| **Our Results** | | | | | | |
| Qwen2.5-0.5B | 0.5B | - | 8.3 | 50.8 | 26.3 | 10.2 |
| SFT | 0.5B | GSM8K | 35.7 | 39.3 | 14.3 | |
| GRPO-clip | 0.5B | GSM8K | 17.6 | 42.8 | 34.3 | 10.3 |
| GRPO-noclip | 0.5B | GSM8K | 40.7 | 68.3 | 53.9 | 17.2 |
| GR-PSPO | 0.5B | GSM8K | 39.7 | 68.1 | 53.2 | 16.8 |
| Qwen2.5-1.5B | 1.5B | - | 1.6 | 2.4 | 1.7 | 5.1 |
| GRPO-clip | 1.5B | GSM8K | 37.8 | 70.9 | 58.4 | 14.9 |
| GRPO-noclip | 1.5B | GSM8K | 57.9 | **80.4** | **74.9** | 25.2 |
| GR-PSPO | 1.5B | GSM8K | **59.4** | 77.7 | 70.3 | **25.6** |

Table 5: Performance of similar-size models from the literature, compared with our results. We report our model accuracy from the zero-shot Top-1 accuracy.
*These models were evaluated against MATH rather than MATH-500. Where MATH-500 is a subset of MATH, we still compare the results here.

# B    LLM-AS-JUDGE SETUP

We use the following rubric:

> You are an evaluator. Judge ONLY the reasoning quality (not final answer). Score four criteria from 0 (bad) to 5 (excellent): 1) Uses given constraints/numbers 2) Logical flow (no leaps/contradictions) 3) Mathematical soundness of steps 4) Clarity/conciseness (limited repetition/hedging) Return strict JSON: "score":"constraints":X,"logic":Y,"math":Z,"clarity":W,"overall":O,"rationale":"...". Overall is the rounded mean of the four sub-scores.

We use the following system prompt:

> You are a fair, strict, and concise evaluator. Output ONLY JSON.

We use the following user prompt:

> Problem: prompt
> Model reasoning: completion
> Provide your JSON now.

We used `llama-3.3-70b-versatile` as our judge model, accessing it via an API service. We averaged scores per dataset per model across all responses and seeds. We used the $T = 0$ answers.

