# OpenReview forum: "It’s Not You, It’s Clipping: A Soft Trust-Region via Probability Smoothing for LLM RL"
_ICLR.cc/2026/Conference — ICLR 2026 Conference Withdrawn Submission_

### Official Review · Reviewer_W7ZF · 2025-10-21

**Soundness:** 2
**Presentation:** 2
**Contribution:** 2
**Rating:** 2
**Confidence:** 3

**Summary:**

The paper presents PSPO a methodology that smooths the importance sampling ratio with respect to the old policy, allowing for gradients to flow even outside the trust region.

**Strengths:**

The paper does give a decent overlay of the problems with clipping and serves as an interesting idea to how possibly more expressive importance ratios could be developed. The general intuition as to how the method clipping method is developed seems interesting specially since there is no additional compute needed for it.

**Weaknesses:**

The paper has a fair amount of weaknesses.

Presentation:

The papers presentation is rather poor. Specifically the mathematical proofs are really hard to read and comprehend, a lot of theory is dropped on the reader at once without any explanation what the point of it is. I read through the theory and more or less understand what it's trying to say but I didn't get anything meaningful out of it. Further it seems like this theory is just laid on the reader for the sake of having theory on the paper, as it does not seem to connect or give intuitions as to how the method was developed from first principles, just gives some bounds and mathematical formulations of trivial things (Lemma 1).


The results seem weak to me, only 1 dataset and smaller models as well as it does not seem to outperform GRPO in out of distribution tasks. Baselines are also weak.

Ablations seem to be missing specifically as the new \alpha parameter was introduced.

Overall I think the paper is a good starting point but needs a fair amount of polishing  and refinement to be a strong contribution. I will list out what I think the paper would greatly benefit from in the questions section.

**Questions:**

I would highly suggest (if they have not done already) the authors to read [1]. This work discusses similar topics of clipping specifically how it relates to off-policy learning and does a great job a guiding the reader through why the pillars of clipping and what is and isn't necessary.

My next request is that the authors compare to [1] as I think they are similar in spirit.

I think the authors need to also compare with Truncated IS.

Additionally, can the authors perform more ablations, specifically on \alpha, as well as produce training on more datasets.




[1] Tapered Off-Policy REINFORCE: Stable and efficient reinforcement learning for LLMs

---

### Official Review · Reviewer_x4oF · 2025-10-21

**Soundness:** 1
**Presentation:** 1
**Contribution:** 1
**Rating:** 0
**Confidence:** 5

**Summary:**

This paper proposes an alternative to the clipping of the importance sampling IS ratio appearing in PPO-like approaches. Specifically, it is proposed to replace the IS ratio $\frac{\pi_\theta(a|s)}{\pi_{old}(a|s)}$ by a smoothed version (loosely and heuristically inspired by label smoothing), corresponding to a ratio $\frac{(1-\alpha)\pi_\theta(a|s) + \alpha \pi_{old}(a|s)}{\pi_{old}(a|s)}$, this ratio being not clipped. A few straightforward but not really informative properties are provided. The resulting algorithm is not really new, it is  Reinforce with importance sampling (and a baseline), scaled by a ration $(1-\alpha)$ that could be incorporated in the learning rate. The resulting approach is experimented on small LLMs, results are mostly on par with a GRPO-noclip baseline (what this baseline is exactly being pretty unclear from the paper).

**Strengths:**

* Clipping the importance sampling ratio is an heuristic, improving it can be impactful.

**Weaknesses:**

(More details are provided below in the “questions” section)
* The quality of writing could be really improved
* The clarity of writing could be improved too. A lot of problem come from apparent (by reading the paper) confusion of the authors regarding what policy gradient is, what importance sampling is, the different kind of regularizations and their reasons, and other related topics.
* The stated theoretical results are correct (there are maths issues though, like undefined objects, etc), they are straightforward, but they provide little to no information about the proposed approach.
* The resulting approach is nothing else than a Reinforce variation  with importance sampling, nothing really new, and something known to be unstable (due to IS, which is a pity, given that stability is one of the initial motivations)
* The proposed approach does not perform better than the noclip baseline, what this baseline is is unclear, and the results of the clip variation are somehow suspicious (is it implemented correctly)

**Questions:**

### Quality of writing
The quality of writing could be improved in a number of ways. Many references are broken (eg, missing years). There is not technical background, and the notations or core concepts are not even introduced (like what is an MDP, what is a state, a policy, in the context of LLMs, etc). There is a series of theoretical results, but no explanation of why they are useful or interesting. There is some effort in providing a reproducible experimental section (hyper parameters, etc), but the baselines are not clear (what is noclip, is clip implemented by the authors or from a library, generally speaking on what library does the experiments build upon, etc). Overall, this could be improved, there is enough room to provide more details and explanations.

### Clarity of writing
There are numerous issues with the clarity of writing. Some examples below.
* There is a reference to “optimal theoretical options” regarding TRPO (l.37), what does it mean?
* There are a number of arguments around the size of stepsizes, being small or big, hindering possibly convergence, this is pretty unclear (knowing well the policy gradient literature), can the authors expand/clarify?
* l.48, “avoid clipping by using a single pass over data”, this is either wrong or badly formulated. One should (theoretically) sample new samples for each batch, not for each epoch, to have an importance ratio equal to 1. From the writing (here and after), it seems that the authors rather consider that a dataset is sampled from $\pi_{old}$, and that a full pass over the generated dataset is performed; after the 1st gradient step/batch, the IS would be in principle different from 1). Please clarify.
* Around l.85, please clarify the role of the different regularization term. The regularization towards $\pi_{ref}$ is part of the problem (eg to avoid reward hacking, to not move to far from a trustable reference policy), while the regularization towards $\pi_{old}$ is for stabilizing updates, essentially for TRPO or PPO to allows for a stable off-policy greedy step in what is fundamentally a policy iteration scheme. Both can be considered in conjonction. Please clarify.
* l.89, “in RL problems there is often multiple optimal actions”. Please justify formally this statement, it is quite debatable.
* It is stated in the paper that the proposed approach “implicitly control the divergence without an explicit KL term” (eg l.198, but other places with similar formulation). Please justify formaly, this is debatable.

### About theoretical results
The results are correct, even if there are maths issues (like $r$ being a function of $\theta$ then of $a$, the term $A$ being not defined, some results being not proven even though straightforward, etc). However, they do not tell much about the proposed approach. For example, there is a lemma about total variation contraction, but why would this be importante. Same thing for the Corollary (bad usage of the word corollary though). What it tells is that your replace $\pi$ by something “close enough” in the importance ratio (depending on $\alpha$), but it tells nothing about how stable the resulting update is.

### Novelty of the approach
Proposition 3 and after states pretty clearly that the proposed approach end up being Reinforce (or the contrastive variant) with importance sampling. IS is well known for introducing a huge variance (that’s why TRPO or PPO exist, in part), and here you just no longer try to tackle the problem (of the possibly exploding variance due to IS). So it is not new, and it does not address the initial motivation. Please clarify if there is a misunderstanding here. Also, section 5 states that “PSPO provides stability without needing to truncate”, can you justify this more formally? This would be really surprising if the understanding of what the method does is correct.

### Experiments and baselines
* What is GRPO-noclip? Is it GRPO without clipping, but with importance ratio, then what is the difference with the proposed approach? Is it GRPO with a sampling ratio of 1? Is it then justified, even doing a single pass over the data, as fresh samples should be generated at each batch (is it what is done)?
* Is GRPO-clip implemented by the authors or from a library ? On what library do the authors build upon ? Do they even at the token level or the sequence level (even deepseek is not crystal clear about this, with different variations in different papers)?
* Can you justify setting $\beta=0$? (Appart from common default)

### Some links to the literature

The authors are encouraged to read the conservative policy iteration (CPI) paper [A], TRPO is a more practical version of it (CPI being cited thoroughly there), and PPO a more practical heuristic approximation of TRPO. This is even more true that CPI consists in mixing policies.


Given that the paper addresses the limitations of clipping, it misses a few approaches from the literature, in the context of LLMs, either proposing an alternative to clipping the importance sampling ratio, or even removing the importance sampling ratio. Notably, the following papers could be discussed, and possibly considered as baselines:
* [B] generalizes clipping by clipping the importance ratio within the gradient rather than within the objective, asymmetrically
* [C] is a policy-gradient approach (in the sense of optimizing for a policy), relying on an additional value network but without any importance sampling, while being off-policy.
* [D,E] is an off-policy policy gradient without importance sampling. Interestingly, it basically states that RLOO [F, G] (roughly GRPO on-policy) can be used safely off-policy, without importance sampling (which may be the GRPO-noclip baseline maybe, that can thus be run on more iterations, not just one).


[A] Approximately Optimal Approximate Reinforcement Learning.
[B] Tapered Off-Policy REINFORCE: Stable and efficient reinforcement learning for LLMs, Le Roux et al, 2025.
[C] Offline Regularised Reinforcement Learning for Large Language Models Alignment, Richmond et al, 2024.
[D] Contrastive Policy Gradient: Aligning LLMs on sequence-level scores in a supervised-friendly fashion, Flet-Berliac et al, 2024.
[E] Command A: An Enterprise-Ready Large Language Model, Cohere, 2025.
[F] Back to Basics: Revisiting REINFORCE Style Optimization for Learning from Human Feedback in LLM, Ahmadian et al., 2024
[G] Buy 4 REINFORCE Samples, Get a Baseline for Free! Kool et al, 2019.

---

### Official Review · Reviewer_aRTV · 2025-10-27

**Soundness:** 4
**Presentation:** 3
**Contribution:** 3
**Rating:** 4
**Confidence:** 4

**Summary:**

The paper proposes Probability Smoothing Policy Optimization (PSPO), a new way to stabilize reinforcement learning for large language models. Instead of clipping policy updates like PPO and GRPO do—which can erase useful gradients—PSPO softly smooths the new policy toward the old one before updating. This creates a kind of soft trust region: updates stay stable but never flatline, so the model keeps learning smoothly. They plug this idea into GRPO, forming GR-PSPO, which works as a drop-in replacement for clipping with no extra cost. In short, PSPO keeps training steady without throwing away valuable learning signals.

**Strengths:**

1. The paper is well-motivated - it starts from a clear and practical limitation of current RL methods (PPO, GRPO): the instability and gradient loss caused by ratio clipping. Their solution, Probability Smoothing Policy Optimization (PSPO), is simple and only requires replacing clipping with a soft smoothing mechanism that preserves gradient flow while maintaining stability.

2. The paper’s theory is clear and convincing. The authors show how probability smoothing keeps the updates stable. They show that this prevents gradients from vanishing. So it will naturally create a soft trust region. This supports their method theoretically.


3. I also like that the method is practical and easy to apply (a drop in) - PSPO can directly replace clipping without adding any extra cost, it also makes training more stable and improves performance, which makes it genuinely useful in real RLHF setups.

**Weaknesses:**

1. Why only math tasks? The paper only tests PSPO on math reasoning with clear right/wrong answers. This makes it hard to know if the method works in more subjective or open-ended RLHF settings. Could it behave differently when rewards are noisy or continuous?

2. What’s the real benefit over GRPO-noclip? The results show almost identical accuracy between GR-PSPO and GRPO-noclip. If the main gain is slightly better response quality, is it really worth adding another hyperparameter (α) to tune?

3. Test at larger scales. The paper cites prior work saying GRPO becomes unstable in bigger or sparse models, yet experiments stop at 1.5B parameters. Showing results on 7B or MoE models would better demonstrate whether PSPO truly fixes those issues.

4. How should α be chosen? The paper uses α = 0.1 from a small grid search but gives no guidance or analysis. How sensitive is performance to this value? Does it transfer across models and tasks, or require new tuning each time?

5. Learning rate differences blur the comparison. Table 1 shows that GRPO-clip uses a learning rate ten times higher than GR-PSPO (5e-6 vs. 5e-7 for the 0.5B model). It’s unclear whether these values were tuned independently or jointly. Without consistent hyperparameter tuning, it’s difficult to tell if the performance gap truly comes from clipping vs. smoothing rather than learning rate differences.

6. Unclear motivation for data reuse. GR-PSPO and GRPO-clip are trained with two iterations, while GRPO-noclip uses only one. The paper doesn’t explain when multiple iterations are actually needed—such as in cases with large batch sizes or sample efficiency limits. Without showing where single-pass training fails, it’s hard to see why the two-iteration setup is the right comparison.

**Questions:**

Please read the weakness section and answer the questions listed there. Thank you!

---

### Official Review · Reviewer_nJys · 2025-10-30

**Soundness:** 2
**Presentation:** 1
**Contribution:** 2
**Rating:** 2
**Confidence:** 4

**Summary:**

This paper proposes Probability Smoothing Policy Optimization (PSPO), a method that replaces ratio clipping in policy gradient algorithms (like PPO/GRPO) by smoothing the current policy towards the old behavior policy before computing the importance sampling ratio. The authors instantiate PSPO within GRPO (dubbed GR-PSPO) and demonstrate good empirical results on mathematical reasoning tasks, showing significant performance gains over clipped GRPO and comparable or superior results to an unclipped baseline, while also improving response quality.

**Strengths:**

1. The core idea is well-motivated.

2. The method is presented clearly.

**Weaknesses:**

1. The experimental setup raises significant concerns about the fairness of the comparisons and the strength of the evidence.

- Unfair Baseline: In Tables 2 and 3, the proposed method is compared against published works that were trained on a mixture of GSM8K and MATH, while the authors' models are trained on GSM8K only. This gives an unfair advantage to the baselines. A fair comparison requires training another model under the same GSM8K+MATH training setting.

- The choice to compare on the subset MATH-500 instead of the full MATH dataset, while comparing to results reported on the full set, makes it difficult to assess the true performance gap.

2. Insufficient Technical Content: The paper appears to be padded with content that does not substantially contribute to the scientific narrative, suggesting a lack of depth.

- Excessive space is dedicated to routine details, such as a table of hyperparameters (Table 1) and verbose footnotes in Tables 2/3 explaining the MATH-500 discrepancy.

- The inclusion of a lengthy, qualitative example in Section 4 and redundant discussion on the base model's performance (Page 9) feels like filler material, especially when the core experimental comparisons are not yet fully justified. This space could be better used for the fair comparisons requested above.

**Questions:**

Please refer to the Weaknesses.

---

### Author Response · Authors · 2025-11-17

We appreciate the reviewers’ time and feedback. After discussing, we have decided to withdraw the paper from ICLR as we will not be able to complete the additional experiments within the rebuttal window. The comments were valuable, and we will incorporate your feedback as we revise the work. Thank you again for your time.

---

### Note · Authors · 2025-11-17

I have read and agree with the venue's withdrawal policy on behalf of myself and my co-authors.